# Network Pharmacology of Ginseng (Part II): The Differential Effects of Red Ginseng and Ginsenoside Rg5 in Cancer and Heart Diseases as Determined by Transcriptomics

**DOI:** 10.3390/ph14101010

**Published:** 2021-09-30

**Authors:** Alexander Panossian, Sara Abdelfatah, Thomas Efferth

**Affiliations:** 1EuroPharma USA Inc., Green Bay, WI 54311, USA; 2Department of Pharmaceutical Biology, Institute of Pharmaceutical and Biomedical Sciences, Johannes Gutenberg University, 55099 Mainz, Germany; saabdelf@uni-mainz.de

**Keywords:** red ginseng HRG80^TM^, ginsenoside rg5, gene expression, IPA pathways, network pharmacology, transcriptomics

## Abstract

*Panax ginseng* C.A.Mey. is an adaptogenic plant traditionally used to enhance mental and physical capacities in cases of weakness, exhaustion, tiredness, or loss of concentration, and during recovery. According to ancient records, red ginseng root preparations enhance longevity with long-term intake. Recent pharmacokinetic studies of ginsenosides in humans and our in vitro study in neuronal cells suggest that ginsenosides are effective when their levels in blood is low—at concentrations from 10^−6^ to 10^−18^ M. In the present study, we compared the effects of red ginseng root preparation HRG80^TM^(HRG) at concentrations from 0.01 to 10,000 ng/mL with effects of white ginseng (WG) and purified ginsenosides Rb1, Rg3, Rg5 and Rk1 on gene expression in isolated hippocampal neurons. The aim of this study was to predict the effects of differently expressed genes on cellular and physiological functions in organismal disorders and diseases. Gene expression profiling was performed by transcriptome-wide mRNA microarray analyses in murine HT22 cells after treatment with ginseng preparations. Ingenuity pathway downstream/upstream analysis (IPA) was performed with datasets of significantly up- or downregulated genes, and expected effects on cellular function and disease were identified by IPA software. Ginsenosides Rb1, Rg3, Rg5, and Rk1 have substantially varied effects on gene expression profiles (signatures) and are different from signatures of HRG and WG. Furthermore, the signature of HRG is changed significantly with dilution from 10,000 to 0.01 ng/mL. Network pharmacological analyses of gene expression profiles showed that HRG exhibits predictable positive effects in neuroinflammation, senescence, apoptosis, and immune response, suggesting beneficial soft-acting effects in cancer, gastrointestinal, and endocrine systems diseases and disorders in a wide range of low concentrations in blood.

## 1. Introduction

*Panax ginseng* C.A.Mey. is likely one of the most widely used botanicals in the world [1,2]. This adaptogenic plant [2,3,4,5,6,7,8] is approved in Europe and other countries as an herbal medicinal product to enhance cognitive functions, physical capacities in weakness, exhaustion, tiredness, loss of concentration, and during convalescence [9,10]. According to ancient records, red ginseng root enhances longevity with long-term intake [11]. Overall, ginseng is a promising treatment for aging-related diseases, including neurodegenerative, cardiovascular diseases, diabetes, and cancer [1,2,3,4,5,6,7,8,9,10,11,12,13,14,15].

A challenge in ginseng research and herbal therapy, in general, is the consistency and reproducibility of the results obtained from various studies [10]. One of the critical solutions to this problem is the reproducible quality of herbal preparations, which is possible to achieve by implementation of GMP and Good Agriculture Practices in the production of herbal preparations, including their cultivation in well-controlled conditions.

Recently, we have demonstrated that both hydroponically cultivated red ginseng root powder HRG80^TM^ (HRG) and white ginseng (WG) preparations were effective in preventing and mitigating the stress-induced deterioration of cognitive functions in healthy subjects [16] and elderly patients with mild cognitive disorders [17] at daily doses of 418 mg HRG (64 mg of total ginsenosides, 1.2 mg Rg5) and 764 mg WG (19.6 mg of total ginsenosides, 0.16 mg Rg5). Both HRG and WG preparations substantially impacted brain activity, affecting various brain regions depending on the mental load during relaxation and cognitive tasks associated with memory, attention, and mental performance. Both ginseng preparations activated electroencephalogram (EEG) spectral powers compared to placebo [17]. The spectral changes in the quantitative EEG induced by HRG indicated an improvement in mood and calming effects evidenced by the modulation of β2 waves, representing changes in GABA-ergic neurotransmission. HRG attenuated δ/θ wave power, which is increased in aging [17]. Red ginseng HRG preparation was more active than WG in humans [17] and in an animal study, where HRG induced higher excitability of pyramidal cells by modulation of ionotropic glutamate NMDA and kainate receptor-mediated transmission at daily doses of 10 to 50 mg/kg body weight corresponding to human doses of 100–500 mg [18].

We had several aims in this study, intending to cover several gaps in ginseng research and specifically in hydroponically cultivated roots used to produce red ginseng preparations. These are related to the following issues:

Numerous studies of purified ginsenosides show their potential efficacy in treating cancer and other diseases [19,20,21,22,23,24,25,26,27,28], suggesting that the total extract or powdered roots have similar or probably better therapeutic effectiveness [23,24,25,26]. However, clinical evidence supporting this suggestion in comparative studies is missing [27,28].

As a matter of fact, blood cells and tissues are continuously exposed to varying concentrations of ginsenosides after oral administration of ginseng in daily therapeutic doses of 0.6–9.0 g [10,29,30]. They range from maximal detected concentrations of 35 ng/mL to the limit of detection 0.5 ng/mL and less within 12–48 h after oral intake [29]. A recent study demonstrated that ginsenoside Rg5 exhibits soft-acting effects in a wide range of physiological and sub-physiological concentrations from 1 μM to 1 aM. However, low-dose studies in corresponding concentrations are not available in the scientific literature.

Pharmacological effects of ginseng preparations at various levels of regulation of homeostasis have been studied [26,31,32,33,34,35,36,37,38]; however, network pharmacology studies [26,33,34,35,36,37,38] at the transcriptomic level of regulation of cellular homeostasis are limited [38].

Therefore, the first aim of this study was to compare the effects of red ginseng root preparation (HRG80^TM^) at concentrations from 0.01 to 10,000 ng/mL with effects of white ginseng (WG) and purified ginsenosides Rb1, Rg3, Rg5, and Rk1 on gene expression of isolated hippocampal neurons. The study’s second aim was to predict the effects of differently expressed genes on cellular and physiological functions in organismal disorders and diseases using a network pharmacology approach to transcriptomics.

## 2. Results

### 2.1. Effect of Ginsenosides Rb1, Rg3, Rg5 and Rk1, WG and HRG80^TM^ at Different Concentrations on Gene Expression Profile in the Hippocampal Neuronal Cell Line HT22

Table 1 shows the number of genes deregulated (>20 fold compared to control) by ginsenosides Rb1, Rg3, Rg5 and Rk1, WG and HRG80^TM^ at different concentrations in the hippocampal neuronal cell line HT22, which in the same range—from 283 to 461 at all tested concentrations of HRG80^TM^ and within 20% RSD of the mean value 383 ± 77 (Appendix A).

The number of genes deregulated by WG and purified ginsenosides is also within this range, despite the difference in the content of ginsenosides Rb1, Rg3, Rg5, and Rk1 in WG and HRG80^TM^ (Table 1). However, the gene expression profile (signature) was substance-specific and concentration-specific, containing many genes that are not deregulated in other concentrations or other test samples alone (Figure 1, Table 1, and Appendix A).

Only one gene, *SUOX*, encoding mitochondrial sulfite oxidase, was commonly deregulated by all purified ginsenosides, WG, and HRG80^TM^ (Figure 1a,b), and the KRTAP 10-7 gene in all tested concentrations of HRG80^TM^ and total extract HRG80^TM^ (Figure 1c).

Deregulated genes expression profiles (signatures) of WG and HRG80^TM^ were also significantly different; only 64 of 398 and 344 genes deregulated by HRG80^TM^ (16% of total) and WG (18% of total), respectively, were the same (Figure 1d).

These observations demonstrate that red ginseng preparation HRG80^TM^ significantly impacts the gene expression of hippocampal neurons in a wide range of concentrations, from 1 μg/mL to 0.01 ng/mL.

These observations also reveal that the gene expression profile of hippocampal neurons is specific for every purified ginsenoside and is quite different from red ginseng HRG80^TM^ and WG preparations.

### 2.2. Effect of Ginsenoside Rg5 on Signaling Canonical Pathways

Figure 2 show the predicted effects (−log *p*-value > 1.3, z-score > 2) of red and white ginseng extracts (HRG80^TM^ and WG) and their major constituents ginsenosides Rb1, Rg3, Rg5, and Rk1 on canonical signaling pathways, including inhibition of:

neurotransmitters and nervous system signaling involved in *neuroinflammation* signaling (Figure 3);*senescence*, *ferroptosis*, * adrenomedullin*, and *WNT/β-catenin* signaling—canonical pathways involved in cellular stress response, injury, and cancer progression (Figure 3 and Figure 4);intracellular second messenger c-AMP mediated *protein kinase A* signaling (Figure 4);transcriptional regulation of *sirtuin* and nuclear receptors, and *estrogen receptor*-mediated signaling (Figure 4).

The effects of ginseng extracts WG and HRG, and ginsenosides Rb1, Rg3, Rg5, and Rk1 on gene expression involved in these signaling pathways are shown in Appendix A in detail.

### 2.3. Predicted Effects of Ginsenosides and Ginseng Extracts HRG80^TM^ and WG

A large proportion (about 75%) of deregulated genes consisted of networks that are significantly associated with cancer, gastrointestinal, and endocrine systems diseases and disorders (Table 2 and Figure 5).

IPA analysis shows diseases and bio functions (Table 3) that are expected to be significantly (activation score z > [±2],−log *p*-value) > 1.5) correlated to ginseng extracts HRG80^TM^ and WG, and ginsenosides Rb1, Rg3, Rg5. HRG80^TM^ is expected to inhibit endocrine gland tumors, abdominal cancer and neoplasm, necrosis, connective tissue cell death, and heart disease at a concentration of 1 μM.

At lower concentrations (100 or 10 ng/mL), HRG80^TM^ may inhibit neoplasia of cells, invasion of breast cancer cell lines, and growth of tumors and endocrine gland tumors, while the highest concentration of 10 μg/mL may inhibit necrosis and proliferation of leukemia cell lines.

WG is expected to inhibit hepatic stenosis and abdominal neoplasm at the same concentration of 10 μg/mL. Antitumor activity profiles of purified ginsenosides are different from ginseng total extracts, where the most polyvalent is Rg5, which is expected to inhibit genital and genitourinary tumors, malignant neoplasm and solid tumors, neoplasia of cells, pelvic tumors, carcinomas, and sarcomas (Table 3 and Figure 5).

Expected inhibition of endocrine gland tumors at two different concentrations of 1 μg/mL and 10 pg/mL (Figure 6) is associated with two distinct networks (Figure 6a,b) including quite different sets of molecules, where 72 genes are commonly deregulated at these concentrations, while 250 and 251 are unique for each of them, respectively (Figure 6c and Appendix A). IPA analysis of gene expression of neurons shows predicted inhibition of endocrine tumors at both a concentration of HRG80 of 10 ng/mL (HRG4) and of 1000 ng/mL (HRG2). The same canonical pathways were inhibited in both concentrations; however, these effects were mediated by quite different sets of deregulated genes. Figure 5 shows only 72 genes involved in inhibiting endocrine tumors deregulated at both concentrations of HRG. In addition, 260 genes contribute to the overall anti-tumor effect at a concentration of 1000 ng/mL, and 251 other genes at a concentration of 10 ng/mL. Since the content of Rg5 comprises 7.534% of the extract, the concentration of Rg5 in cell culture incubation media is 75.3 ng/mL (98 nM) and 0.75 ng/mL (0.98 nM), respectively (Table 1). This is in line with the results of pharmacokinetic studies in humans, where the max level of Rg5 in blood was found to be 8 ng/mL.

Similarly, expected inhibition of abdominal neoplasm by red HRG2 and white WG ginseng is associated with two distinct networks (Figure 7a,b) including quite different sets of molecules, where 54 genes are commonly deregulated by HRG or WG, while 407 and 290 are unique for each ginseng preparation, respectively (Figure 7c).

## 3. Discussion

This study’s primary aim was to utilize transcriptomics of neuronal cells to uncover potential pharmacological activities and indications for use in medicine hydroponically cultivated Panax ginseng root preparation HRG80^TM^ (HRG). This task is closely associated with exploring HRG at reasonable in vitro concentrations matching the concentrations of ginsenosides found in in vivo studies on human subjects.

Our study provides evidence that HRG is pharmacologically active at the concentrations 10, 100, 1000, and 10,000 ng/mL, corresponding to the concentrations of ginsenosides (from 0.05 to 35 nM, equivalent to 5–7000 ng/mL, assuming that the content of ginsenoside Rg5 is 2% in the HRG80^TM^, Appendix A) detected in the blood of human subjects orally administered with red ginseng preparations in therapeutic doses [29].

At all tested concentrations, HRG deregulated genes significantly associated with cancer, gastrointestinal, and endocrine system diseases and disorders (Table 2 and Figure 4).

However, the expected (based on IPA analysis results) pharmacological profile and possible indications for use are different for both WG and HRG, as well as for various doses of HRG in blood. HRG80^TM^ has the potential to inhibit endocrine gland tumors, abdominal cancer and neoplasm, necrosis, connective tissue cell death, and heart disease at blood concentrations of 1 μM, while at other concentrations, HRG80^TM^ has the potential to inhibit neoplasia of cells, invasion of breast cancer cell lines, growth of tumors and endocrine gland tumors, and necrosis and proliferation of leukemia cell lines. On the other hand, WG can presumably inhibit hepatic stenosis, which is not expected with HRG.

These conclusions are in line with other reports on the antitumor activity of ginseng and purified ginsenosides [19,23,25,26,36,39,40,41,42,43,44].

Meanwhile, antitumor activity profiles of purified ginsenosides are different from ginseng total extracts (Table 3). The results obtained for purified ginsenosides cannot be simply extrapolated on the total extract or powdered roots, which exhibit quite different pharmacological and therapeutic activities.

The mechanism of action of HRG80^TM^ is multitargeted and associated with stress-induced abnormalities, neuroinflammation, and senescence, and other signaling pathways (Figure 3, Figure 4, Figure 5 and Figure 6) playing an important role in these physiological and cellular processes, including apoptosis, tumorigenesis, and progression of cancer, which is typical for adaptogens [8,45,46].

Overall, the results of this study are in line both with the traditional use of ginseng in aging and the theory in which low-grade chronic inflammation (inflammaging) that develops with senescence plays a crucial role in the progression of aging-related diseases—primarily cancer [47,48,49]—, and in line with the adaptogen concept, suggesting efficacy of adaptogens in stress-induced aging-related diseases [8,50,51] (Figure 8).

There are several constraints in this study. One of them is the lack of scientific literature about the direction (positive or negative) of correlations between gene expression and physiological function or disease for predicting effects of some experimental findings used in silico analysis. The second is related to the number of concentration points in the dose–response correlation study; more intermediate points over the 10-fold difference range will show smooth changes from point to point. Based on the results of the IPA analysis, we do not have yet explanations for observations of differential regulation of gene expression profiles at different concentrations of ginseng extracts or individual ginsenosides. More mechanistic studies at multiple concentration points within narrower concentration ranges are required to elucidate the regulatory mechanisms, which, clearly, are multi-targeted, multilevel, upstream, and downstream feedback-regulated, resulting in maintaining of intracellular and extracellular homeostasis.

Overall, the effects of total extracts on gene expression profiles are different from the effects of purified ginsenosides in all tested concentrations; however, the predicted effects on signaling pathways, biological functions, and diseases are similar, probably due to adaptive compensatory regulation of alternative signaling molecules of molecular networks.

Further preclinical and clinical studies are required to assess the therapeutic efficacy of HRG80^TM^ in different types of endocrine tumors and abdominal cancers.

Overall, this is the first evidence of pharmacological activity of ginseng preparations at concentrations found in the blood of human subjects after oral administration in therapeutic doses.

## 4. Materials and Methods

All materials and methods used in the present study have been described in detail in our previously published studies [52,53]. Therefore, only a short description of herbal extracts, mRNA microarray hybridization, and ingenuity pathway analysis (IPA) is provided below.

### 4.1. Test Samples and Reference Standard

Powdered red ginseng preparation HRG80^TM^ and its extract (HRG) were obtained at Botalys S. A. (Ath, Belgium). Harvested roots were air-dried and steamed to red ginseng. The red ginseng HRG80^TM^ preparation was standardized for the content of the ginsenosides Rg1, Re, Rf, Rb1, Rg2, Rc, Rh1, Rb2, F1, Rd, Rg6, F2, Rh4, Rg3-(S-R), PPT (20-R), Rk1, C(k), Rg5, Rh2, Rh3, 20S-PPT, and PPD (Appendix A).

Powdered red ginseng preparation HRG80^TM^ and its extract (HRG) were obtained at Botalys S. A. (Ath, Belgium). Korean Ginseng (*P. ginseng* Meyer) root was hydroponically cultivated in controlled conditions, air-dried, steamed to red ginseng, which was powdered, and standardized for the content of ginsenosides Rg1, Re, Rf, Rb1, Rg2, Rc, Rh1, Rb2, F1, Rd, Rg6, F2, Rh4, Rg3-(S-R), PPT (20-R), Rk1, C(k), Rg5, Rh2, Rh3, 20S-PPT, and PPD to obtain red ginseng HRG80^TM^ preparation containing 7.6% of total ginsenosides (Appendix A). HRG80^TM^ preparation was exhaustively extracted by 40% ethanol and evaporated to dryness to obtain HRG extract (DER 4: 1) used for in vitro experiments. The content of ginsenosides in the HRG extract was 30.32% (Appendix A).

The reference standard, *P. ginseng* Meyer powdered root preparation, and the extract contained 5.57% and 22.15% total ginsenosides, respectively (Appendix A). All herbal preparations were analyzed and certified by Botalys S. A.

Purified reference standards of ginsenosides were purchased from Merck https://www.sigmaaldrich.com/, accessed on 27 September 2021.

### 4.2. mRNA Microarray Hybridization

After seeding, we let hippocampal neuronal HT22 cells of mouse origin grow for 24 h before using them for treatment with test compounds. Following this, the cells were subjected to different drug concentrations and combinations or DMSO as a solvent control (0.5%) for another 24 h. An InviTrap^®^ Spin Universal RNA mini kit (250 preps; Stratec Molecular) was used for RNA isolation. A total RNA Nanochip assay was applied on an Agilent 2100 bioanalyzer (Agilent Technologies GmbH) for quality control of RNA, and an RNA index value of >8.5 served as a threshold for further processing of RNA samples. For every two independent experiments, quality control of RNA was performed. The mRNA microarray hybridization was performed at the Genomics and Proteomics Core Facility, German Cancer Research Center (Heidelberg, Germany) using Affymetrix GeneChips^®^ with mouse Clariom S assays. The measurements were normalized by a quantile normalization algorithm without background subtraction. The standard deviation differences were calculated in one-by-one comparisons to identify differentially regulated genes. Chipster software (The Finnish IT Center for Science CSC) was used for further evaluation of results.

### 4.3. Ingenuity Pathway Analysis (IPA)

The interpretation of microarray data and gene expression changes was performed using ingenuity pathway analysis (IPA) software, summer release 2021 (QIAGEN Bioinfor-matics, Aarhus C, Denmark), which relies on the Ingenuity Knowledge Base, a continuously updated database that gathers research findings, with more than 8.1 million findings manually curated from the biomedical literature or integrated from 45 third-party databases. The IPA network contains 40,000 nodes representing mammalian genes, molecules, and biologic functions, linked by 1,480,000 edges representing experimentally observed cause–effect relationships (either activating or inhibiting) related to gene expression, transcription, activation, molecular metabolism, and binding [54].

Using the IPA Core Analysis tool for all tested transcriptomic datasets, we performed predictive analyses of the impact of test samples on canonical signaling and metabolic pathways, which displayed the molecules of interest within well-established pathways; and diseases, disorders, molecular and cellular functions that are activated or inhibited downstream and upstream of the genes, whose expression has been altered.

### 4.4. Statistical Analysis

Two statistical methods of analysis of gene expression datasets were used in IPA: (i) the gene-set-enrichment method, where differentially expressed genes are intersected with sets of genes that are associated with a particular biological function or pathway, providing an ‘enrichment’ score (Fisher’s exact test p-value) that measures overlap of observed and predicted regulated gene sets [55,56], (ii) a method based on cause–effect relationships related to the direction of effects reported in the literature, that provides a so-called z-score that measures the match of observed and predicted up/down-regulation [55,56,57,58]. The predicted effects are based on gene expression changes in the experimental samples relative to the control; z-score > 2, -log *p*-value > 1.3.

## 5. Conclusions

In this study, we, for the first time, have demonstrated that the gene expression profile of murine hippocampal neuronal HT22 cells is changed significantly in response to exposure of hydroponically cultivated red ginseng preparation HRG80^TM^ in concentrations ranging from 10 μg/mL to 0.01 ng/mL. This is in line with concentrations of ginsenosides Rb1, Rg3, Rg5, and Rk1 found in in vivo studies of human subjects taking orally therapeutic doses of red ginseng. Purified ginsenosides Rb1, Rg3, Rg5, and Rk1 have substantially varied effects on gene expression profiles (signatures) and predicted pharmacological activities, as determined by in silico analysis of transcriptomics. Their signatures are different from the signatures and expected therapeutic indications of red ginseng HRG80^TM^ and white ginseng preparations, which are also quite varied.

Comparative in silico transcriptome analysis of microarray-based gene expression profiles of neuronal cells exposed to red and white ginseng extracts and their major constituents ginsenosides Rb1, Rg3, Rg5, and Rk1 predicts a potential beneficial effect in neuroinflammation, senescence, and cancer, gastrointestinal, and endocrine system diseases and disorders. HRG80^TM^ has the potential to inhibit endocrine gland tumors, abdominal cancers and neoplasm, necrosis, connective tissue cell death, and heart disease at blood concentrations of 1 μg/mL.

## Figures and Tables

**Figure 1 pharmaceuticals-14-01010-f001:**
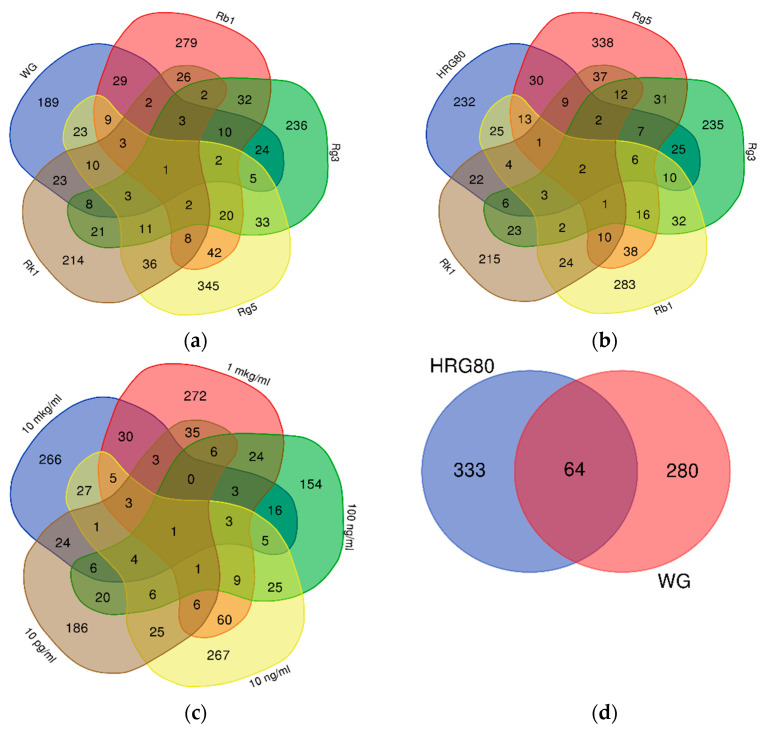
Venn diagrams of: (**a**) the number of genes deregulated by WG and ginsenosides Rb1, Rg3, Rg5 and Rk1, (**b**) the number of genes deregulated by HRG and ginsenosides Rb1, Rg3, Rg5 and Rk1, (**c**) the number of genes deregulated by HRG at concentrations of 10 μg/mL, 1 μg/mL, 100 ng/mL, 10 ng/mL, and 0.01 ng/mL; (**d**) the number of genes deregulated by WG and HRG in the hippocampal neuronal cell line HT22.

**Figure 2 pharmaceuticals-14-01010-f002:**
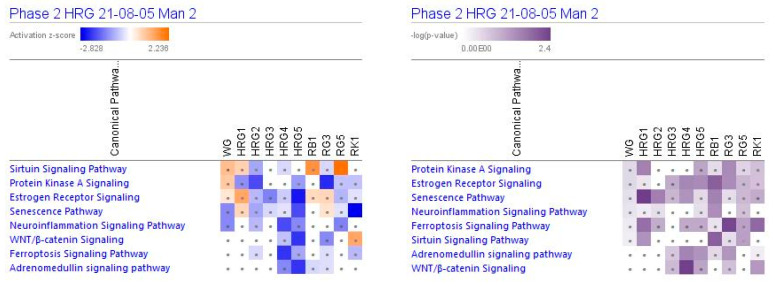
Effects of ginseng extracts WG (10 μg/mL), HRG at concentrations of 10 μg/mL (HRG1), 1 μg/mL (HRG2), 100 ng/mL (HRG3), 10 ng/mL (HRG4), and 10 pg/mL (HRG5), and ginsenosides Rb1, Rg3, Rg5, Rk1 at a concentration 100 nM on selected canonical pathways. The brown color shows the predicted activation and the blue color the predicted inhibition of diseases; symbol · shows that the activation z-score was < 2 and a -log *p*-value < 1.3 = *p* < 0.05. An absolute z-score ≥ 2 is considered significant activation (+) or inhibition (−). The activation z-score predicts the activation state of the canonical pathway, using the gene expression patterns of the genes within the pathway. The *p*-value calculated using a right-tailed Fisher’s exact test indicates the statistical significance of the overlap of analyzed dataset genes that are within a pathway.

**Figure 3 pharmaceuticals-14-01010-f003:**
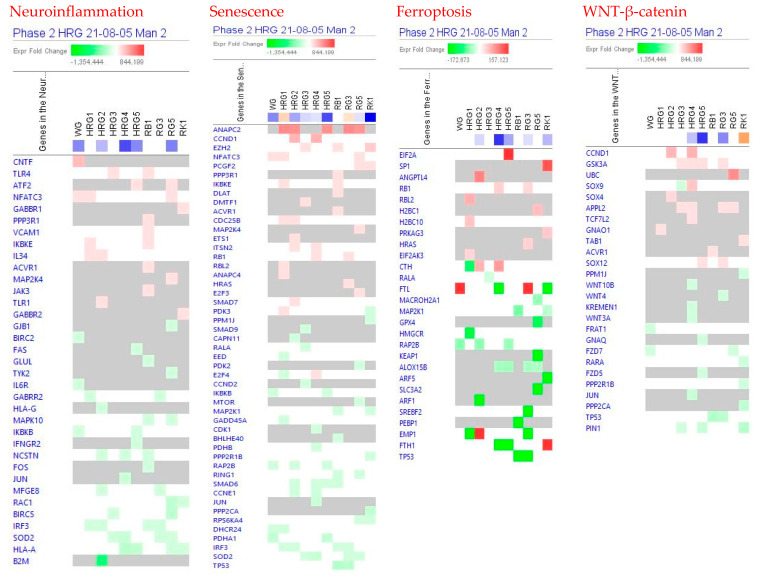
Effects of WG at a concentration of 10 μg/mL, HRG at concentrations of 10 μg/mL (HRG1), 1 μg/mL (HRG2), 100 ng/mL (HRG3), 10 ng/mL (HRG4), and 10 pg/mL (HRG5), and ginsenosides Rb1, Rg3, Rg5 and Rk1 at a concentration of 100 nM on neuroinflammation, senescence, ferroptosis, and WNT—β-catenin signaling pathways. The heatmap of gene expression (in fold-changes compared to controls; red—upregulation, green—downregulation), after exposure of cells with test samples at different concentrations; the column represents signatures of test samples with solid red or green squares indicating genes that are expected to be upregulated or downregulated, respectively; color intensity indicates the actual log-fold changes.

**Figure 4 pharmaceuticals-14-01010-f004:**
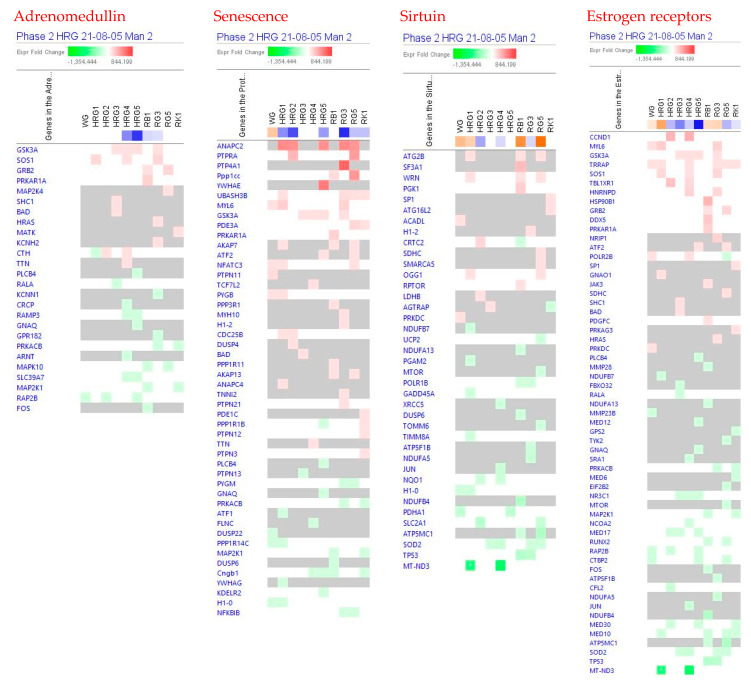
Effects of WG at a concentration of 10 μg/mL, HRG at concentrations of 10 μg/mL (HRG1), 1 μg/mL (HRG2), 100 ng/mL (HRG3), 10 ng/mL (HRG4), and 10 pg/mL (HRG5), and ginsenosides Rb1, Rg3, Rg5 and Rk1 at a concentration of 100 nM on adrenomedullin, PKA, sirtuin and estrogen receptor-signaling pathways. The heatmap of gene expression (in fold-changes compared to control, red—upregulation, green—downregulation), after exposure of cells with test samples at different concentrations; the column represents signatures of test samples with solid red or green squares indicating genes that are expected to be upregulated or downregulated, respectively; color intensity indicates the actual log-fold changes.

**Figure 5 pharmaceuticals-14-01010-f005:**
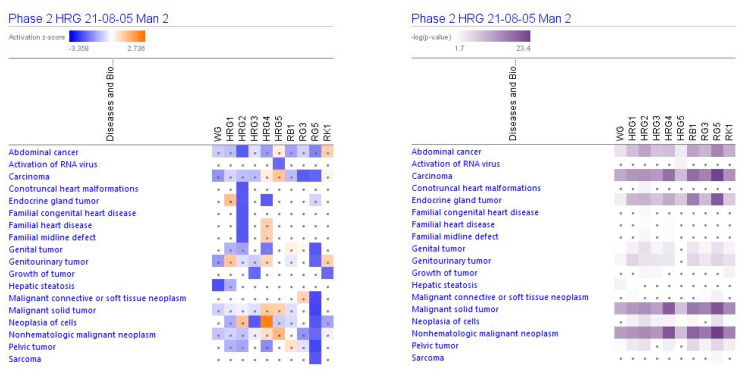
Effects of ginseng extracts WG (10 μg/mL), HRG80^TM^ at concentrations of 10 μg/mL (HRG1), 1 μg/mL (HRG2), 100 ng/mL (HRG3), 10 ng/mL (HRG4), and 10 pg/mL (HRG5), and ginsenosides Rb1, Rg3, Rg5, Rk1 at a concentration 100 nM on diseases. Disease scores are presented using a gradient from dark blue to brown for predicted activation and light to dark blue for predicted inhibition of diseases; symbol **·** shows that the activation z-score was <2 and a -log *p*-value < 1.3 = *p* < 0.05. An absolute z-score ≥ 2 is considered significant activation (+) or inhibition (−).

**Figure 6 pharmaceuticals-14-01010-f006:**
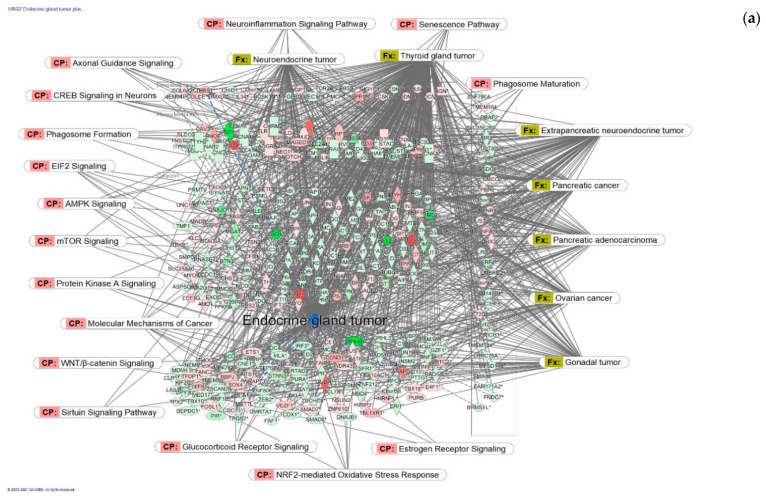
Molecular network shows predicted inhibition of exocrine gland tumor by HRG80^TM^ at concentrations of: (**a**) 1 μg/mL and (**b**) 10 pg/mL. Solid red or green color nodes indicate genes that are up-regulated and down-regulated, respectively; color intensity indicates the actual log-fold changes. The tags labeled with purple display the canonical pathways related to particular genes of the network. The tags labeled with khaki show various types of tumors associated with the molecules in subnetworks. (**c**) Venn diagram showing the number of concentration-specific and commonly deregulated (72) genes at concentrations of 1 μg/mL and 10 pg/mL.

**Figure 7 pharmaceuticals-14-01010-f007:**
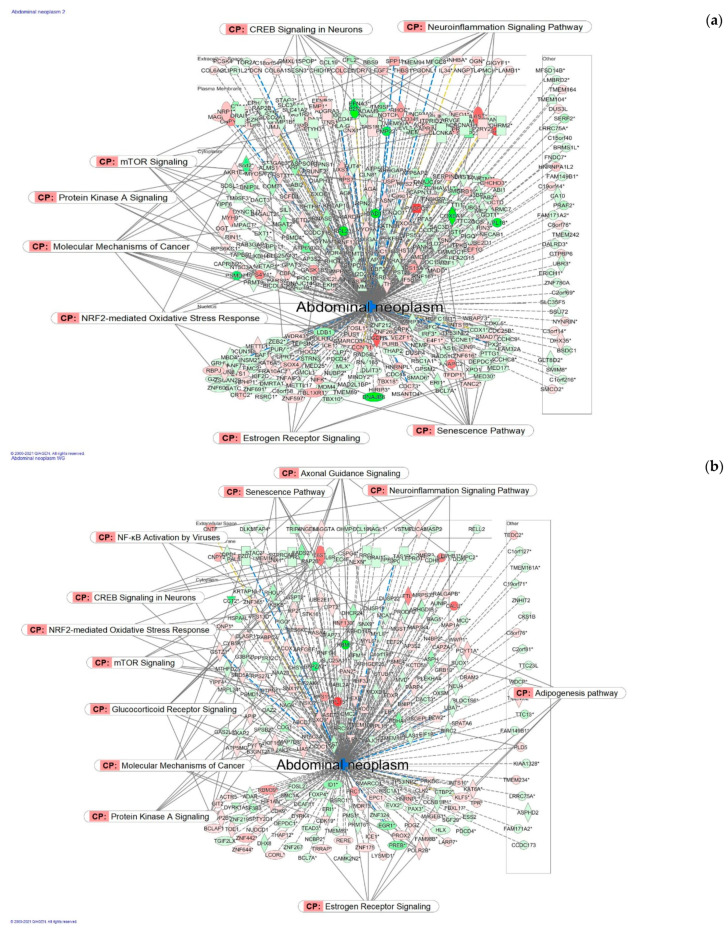
Molecular network shows predicted inhibition of abdominal neoplasm by: (**a**) HRG80^TM^ at a concentration of 1 μg/mL and (**b**) WG at a concentration of 10 μg/mL. Solid red or green color nodes indicate upregulated and downregulated genes, respectively; color intensity indicates the actual log-fold changes. The tags labeled with purple display the canonical pathways related to particular genes of the network. The tags labeled with khaki show various types of tumors associated with the molecules in subnetworks. (**c**) Venn diagram showing the number of product-specific and commonly deregulated (54) genes.

**Figure 8 pharmaceuticals-14-01010-f008:**
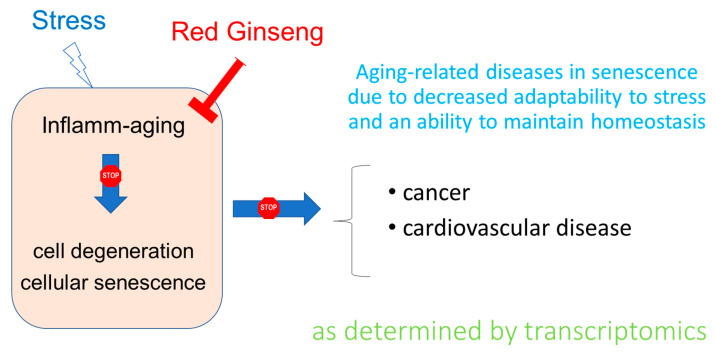
Hypothetical scheme of the effect of Ginseng on aging-related diseases as determined by transcriptomics.

**Table 1 pharmaceuticals-14-01010-t001:** The number of genes deregulated (>20 fold compared to controls) by ginsenosides Rb1, Rg3, Rg5 and Rk1, WG and HRG80^TM^ at different concentrations in the hippocampal neuronal cell line HT22.

Sample Name	Concentration ng/mL; nM	Concentration of Ginsenoside Rg5 nM	Content in WG Extract/Powder, %	Content in HRG80^TM^ Extract/Powder %	Number of Deregulated Genes	Substance/ Concentration Specific Genes
WG	10,000	4.04	n/a	n/a	344	189
HRG1	10,000	982.27	n/a	n/a	397	232
HRG2	1000	98.17	n/a	n/a	461	397
HRG3	100	9.78	n/a	n/a	283	246
HRG4	10	0.91	n/a	n/a	448	290
HRG5	0.01	0.01	n/a	n/a	327	169
Rb1	110.9; 100	-	4.250/1.069	0.183/0.046	470	283 */279 **
Rg3	78.5; 100	-	0.309/0.080	4.307/1.080	413	235 */236 **
Rg5	76.7; 100	100	0.031/0.008	7.534/1.888	553	338 */345 **
Rk1	76.7; 100	-	0.112/0.028	4.027/1.009	373	215 */214 **

*—compared to WG, **—compared to HRG1.

**Table 2 pharmaceuticals-14-01010-t002:** The number of genes associated with various diseases and deregulated by ginseng extracts HRG80^TM^ and WG, and ginsenosides Rb1, Rg3, Rg5.

Test Substance	WG	HRG-80^TM^					Rb1	Rg3	Rg5	Rk1
Concentration, ng/mL	10,000	10,000	1000	100	10	0.01	100 nM	100 nM	100 nM	100 nM
**Diseases and disorders**										
Cancer	300	346	412	253	404	277	419	363	494	338
Gastrointestinal	259	308	360	222	357	243	370	317	439	301
Endocrine System	227	293	336	205	342	228	356	292	430	268
Neurological				167			265			
Reproductive System		237								
Hepatic									255	
Hematological					136					
Hereditary			116							
Infectious								77		
Developmental	68									
Metabolic						9				
Ophthalmic										9
Total genes	344	397	461	283	448	327	470	413	553	373

**Table 3 pharmaceuticals-14-01010-t003:** Predicted effects of ginseng extracts HRG80^TM^ and WG, and ginsenosides Rb1, Rg3, Rg5 in various diseases and biofunctions expressed by activation z-score values. Positive values show predicted activation; negative values showpredicted inhibition.

Test Substance	WG	HRG-80^TM^					Rb1	Rg3	Rg5	Rk1
Concentration, ng/mL	10,000	10,000	1000	100	10	0.01	100 nM	100 nM	100 nM	100 nM
Diseases and Biofunctions										
Abdominal cancer/Abdominal neoplasm	−0.7 −2.062 *	−0.867	−2.162 * −2.018 *	−0.447	−1.348	0.342	−1.274	−0.63	−1.664	0.922
Activation of RNA virus						−2.0 *				
Carcinoma	−1.437	−0.583	−0.849	−0.897	0.331	1.103	−0.92	−2.16 *	−2.042 *	0.206
Colorectal cancer cell viability										−2.009 *
Conotruncal heart malformations Connective tissue cell death			−2.2 * −2.243 *							
Endocrine gland tumor		1.172	−2.176 *		−2.186 *				−0.594	
Familial congenital heart disease			−2.2 *							
Familial heart disease			−2.2 *		0.9					
Familial midline defect			−2.2 *		0.9					
Genital tumor		−1.039	−1.262		−1.838		0.3	0.135	−2.207 *	
Genitourinary tumor	−1.387	1.078	−0.444	−0.699	0.786		−0.319	−0.008	−2.062 *	0.875
Growth of tumor			0.048	−2.012 *						−2.001
Hepatic steatosis	−2.313 *	−1.102								
Invasion of breast cancer cell lines				−2.014 *						
Malignant neoplasm									−2.002 *	
Malignant solid tumor	−0.639	−0.322	−0.219	−0.452	0.99	0.904	−0.438	0.013	−2.458 *	0.039
Migration of tumor cells										−3.101*
Necrosis		−2.263 *	−2.594 *				−2.304 *			
Neoplasia of cells		−1.108	1.18	−2.195 *	2.348 *	−0.7	−0.505		−2.237 *	−1.277
Nonhematologic malignant neoplasm	−0.799	−0.609	−0.821	−0.255	0.63	1.366	−0.087	−1.432	−2.002 *	−0.204
Pelvic tumor		−1.039	−1.191		−1.522		0.573	−0.417	−2.415 *	
Proliferation of leukemia cell lines		−2.070 *								
Sarcoma									−2.2 *	

*—statistically significant effect, z-score > 2 and -log *p*-value > 1.3.

## Data Availability

Data is contained in the article and Appendix A.

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
