# Peer review of "Network Pharmacology of Ginseng (Part II): The Differential Effects of Red Ginseng and Ginsenoside Rg5 in Cancer and Heart Diseases as Determined by Transcriptomics"

_pharmaceuticals, 2021, doi:10.3390/ph14101010_

Round 1
Reviewer 1 Report
This manuscript demonstrated that the gene expression profile of murine hippocampal neuronal HT22 cells is changed significantly in response to ginseng preparations, including white ginseng and Red Ginseng preparation HRG80 TM in concentrations ranging from 10 μg/ml to 0.01 ng/ml as well as purified ginsenosides Rb1, Rg3, Rg5, and Rk1. The results intended to provide a linkage to the concentrations of ginsenosides found in vivo studies on human subjects. The scientific merit is technically sound, and thus the manuscript is recommended for publication after some revision listed below:
- Both Red Ginseng preparation HRG80 TM and white ginseng contain ginsenosides, of course, with various types and contents. In Discussion (Lines 458-462), the authors pointed out differentially pharmacological activities for Red Ginseng preparation HRG80 TM and white ginseng. Can the authors come up with some explanation or discussion according to the effects on gene expression profiles (signatures) and predicted pharmacological activities as determined by in silico analysis of transcriptomics in this study?
- Why did you use hippocampal neuronal cell line HT22 in the first aim of the study? Please provide some describe.
- As described in Lanes 73-74, ginsenoside Rg5 exhibits soft-acting effects in a wide range of physiological and sub-physiological concentrations from 1 μM to 1 aM. Is the data observed in this study in agreement with the effectiveness of ginsenoside Rg5 in the extremely low concentration (1 aM)?
- Please explain why HRG2 compared with HRG4 in Fig.5.
- Please add “a, b, and c” to the corresponding panel in Figure 6.
Reviewer 2 Report
The authors did a good work from an experimental point of view and I recommend you accept the article for publication in Pharmaceuticals journal.
More specific:
Why was the data highlighted in green in Table 3?
In line with Part 1, the paper paves the way for science.
